# Identification of Genetic Risk Factors of Severe COVID-19 Using Extensive Phenotypic Data: A Proof-of-Concept Study in a Cohort of Russian Patients

**DOI:** 10.3390/genes13030534

**Published:** 2022-03-17

**Authors:** Sergey G. Shcherbak, Anton I. Changalidi, Yury A. Barbitoff, Anna Yu. Anisenkova, Sergei V. Mosenko, Zakhar P. Asaulenko, Victoria V. Tsay, Dmitrii E. Polev, Roman S. Kalinin, Yuri A. Eismont, Andrey S. Glotov, Evgeny Y. Garbuzov, Alexander N. Chernov, Olga A. Klitsenko, Mikhail O. Ushakov, Anton E. Shikov, Stanislav P. Urazov, Vladislav S. Baranov, Oleg S. Glotov

**Affiliations:** 1St. Petersburg State University, 199034 St. Petersburg, Russia; sgsherbak@mail.ru (S.G.S.); anton.chana@gmail.com (A.I.C.); barbitoff@bioinf.me (Y.A.B.); anna_anisenkova@list.ru (A.Y.A.); neurologist@mail.ru (S.V.M.); olkl@yandex.ru (O.A.K.); 2City Hospital No. 40, 197706 St. Petersburg, Russia; b40@zdrav.spb.ru (Z.P.A.); viktoriya14054@gmail.com (V.V.T.); brantoza@gmail.com (D.E.P.); pancu43@gmail.com (R.S.K.); y-eis@inbox.ru (Y.A.E.); eugarbouzov@mail.ru (E.Y.G.); al.chernov@mail.ru (A.N.C.); antonshikov96@gmail.com (A.E.S.); urasta@list.ru (S.P.U.); 3Bioinformatics Institute, 197342 St. Petersburg, Russia; 4Faculty of Software Engineering and Computer Systems, ITMO University, 197101 St. Petersburg, Russia; 5D.O. Ott Research Institute of Obstetrics, Gynaecology, and Reproductology, 199034 St. Petersburg, Russia; misha.grizzli@gmail.com (M.O.U.); vsbar40@mail.ru (V.S.B.); 6North-Western State Medical University Named after Ivan Ivanovich (I.I.) Mechnikov, 195067 St. Petersburg, Russia; 7Children’s Scientific and Clinical Center for Infectious Diseases of the Federal Medical and Biological Agency, 197022 St. Petersburg, Russia; 8Institute of Experimental Medicine, 197376 St. Petersburg, Russia; 9Laboratory for Proteomics of Supra-Organismal Systems, All-Russia Research Institute for Agricultural Microbiology (ARRIAM), 196608 St. Petersburg, Russia

**Keywords:** COVID-19, GWAS, genetic variants, deep phenotyping, NGS, severity, genetic associations

## Abstract

The COVID-19 pandemic has drawn the attention of many researchers to the interaction between pathogen and host genomes. Over the last two years, numerous studies have been conducted to identify the genetic risk factors that predict COVID-19 severity and outcome. However, such an analysis might be complicated in cohorts of limited size and/or in case of limited breadth of genome coverage. In this work, we tried to circumvent these challenges by searching for candidate genes and genetic variants associated with a variety of quantitative and binary traits in a cohort of 840 COVID-19 patients from Russia. While we found no gene- or pathway-level associations with the disease severity and outcome, we discovered eleven independent candidate loci associated with quantitative traits in COVID-19 patients. Out of these, the most significant associations correspond to rs1651553 in *MYH14*
*p* = 1.4 × 10^−7^), rs11243705 in *SETX* (*p* = 8.2 × 10^−6^), and rs16885 in *ATXN1* (*p* = 1.3 × 10^−5^). One of the identified variants, rs33985936 in *SCN11A*, was successfully replicated in an independent study, and three of the variants were found to be associated with blood-related quantitative traits according to the UK Biobank data (rs33985936 in *SCN11A*, rs16885 in *ATXN1*, and rs4747194 in *CDH23*). Moreover, we show that a risk score based on these variants can predict the severity and outcome of hospitalization in our cohort of patients. Given these findings, we believe that our work may serve as proof-of-concept study demonstrating the utility of quantitative traits and extensive phenotyping for identification of genetic risk factors of severe COVID-19.

## 1. Introduction

At the end of 2019, a pneumonia outbreak was reported in Wuhan, China. It was caused by a new strain of coronavirus, severe acute respiratory syndrome coronavirus 2 (SARS-CoV-2), and was later named COVID-19 [1]. Up to January 2022, over 340 million individuals were affected, including more than 5.5 million deaths (World Health Organization, https://covid19.who.int/, accessed 24 January 2022).

For two years, researchers have been trying to understand the association between the various symptoms of the disease and host genetics. Identifying specific single nucleotide polymorphisms (SNPs) and other genome variations associated with COVID-19 severity and other symptoms is very important as it can help scientists and clinicians better understand disease pathology. Prediction of genetic susceptibility to COVID-19 might aid clinicians to choose the right treatments for patients (e.g., in a recent study of Indian populations [2]).

Recent studies reported several dozen associations between genetic variants and morbidity, severity, mortality of COVID-19 among different ethnicities (reviewed in Suh et al. [3]). For example, one genome-wide association study (GWAS) conducted in the United Arab Emirates on a sample of 600 participants identified eight susceptibility loci for severe COVID-19. Genes at these loci were found to be linked with T-cell-mediated inflammation and the production of inflammatory cytokines [4]. Another study on a European cohort (seven hospitals from Italy and Spain) detected two cross-replicating associations of severe COVID-19 with variants at the 3p21.31 and 9q34.2 loci [5]. These loci span multiple genes, including *SLC6A20*, *LZTFL1*, *CCR9*, *FYCO1*, *CXCR6*, *XCR1 for 3p21.31* and *ABO*—for 9q34.2.

Despite the large number of reported associations, variants that were discovered in one study can be insignificant in another [3]. To identify loci that show association across cohorts and ethnicities, large-scale meta-analyses are being conducted, including the COVID-19 HG project [6]. The latest meta-analysis results released by the COVID-19 HG include 23 susceptibility loci with great significance. The loci identified in this meta-analysis include the aforementioned 3p21.31 and 9q34.2, as well as several other important ones. While many of the loci span immunity-related genes, the top associations identified at 6p21.1, 12q24.13, and 21q22.11 also have a significant effect on gene expression in the lung.

While genome-wide studies have been generally successful in identifying genetic risk factors of COVID-19, the search for new associations in poorly studied populations might be complicated due to limited cohort size and/or in case of limited breadth of genome coverage (i.e., in studies based on WES or targeted sequencing). Recently, we used targeted sequencing to analyze variants associated with COVID-19 severity in Russia within the *ACE2* gene [7]. In this study, we set off to identify additional susceptibility loci associated with severe COVID-19 using available clinical exome sequencing data. To do so, we leveraged extensive phenotypic data for a cohort of 840 Russian COVID-19 patients.

## 2. Materials and Methods

### 2.1. Study Design and Inclusion Criteria

The study design is an observational clinical trial. We utilized 840 medical records from COVID-19 patients who received treatment in the St. Petersburg State Budgetary Institution of Healthcare City Hospital 40 (City Hospital 40, St. Petersburg, Russia) from 18 April 2020 to 21 November 2020. The size of the sample is expected to provide sufficient power for the identification of quantiative trait loci (QTL) according to recent studies [8]. The patients tested positive for SARS-CoV-2 RNA by polymerase chain reaction (PCR) amplification of nucleic acids from clinical material and presented clinical manifestations and symptoms (fever, general fatigue, malaise, cough, and dyspnea), features of viral pneumonia seen on unenhanced lung CT scan (noted as multiple lobular abnormalities often located in the peripheral areas of the lower lobes and manifested with predominantly perivascular bilateral disease distribution; multiple peripheral areas of ground-glass opacities with rounded morphology and variable extent; interlobular septal thickening/flattening that causes a crazy-paving pattern, areas of consolidation, air bronchogram sign, etc.) [9,10].

### 2.2. Characteristics of Groups of Patients

In accordance with the International and Russian Recommendations for the Prevention, Diagnosis and Treatment of New Coronavirus Infection (COVID-19), all patients were divided in three groups of comparable age ([11]; Ministry of Health of the Russian Federation, 2020). The three groups corresponded to patients with a mild (49 patients, 5.8%), moderately severe (436, 51.9%), and severe (or extremely severe) (355, 42.2%) course of disease. The criteria for a mild course were considered to be body temperature below 38 °C, cough, weakness, sore throat, and the absence of criteria for moderate and severe courses. The criteria for a moderate course are fever, temperature above 38 °C, respiratory rate over 22/min, dyspnea, pneumonia (exposed to CT of the lungs), and SpO_2_ < 95%. Clinical and radiological criteria for severe course were respiratory rate more than 30/min, SpO_2_ ≤ 93%, PaO_2_/FiO_2_ ≤ 300 mmHg, progression of changes in the lungs typical for COVID-19 pneumonia according to CT data, including an increase in the prevalence of revealed changes by more than 25%, as well as the appearance of signs of other pathological conditions, changes in the level of consciousness, unstable hemodynamics (systolic blood pressure less than 90 mmHg or diastolic blood pressure less than 60 mmHg, urine output less than 20 mL/h), and qSOFA > 2 points. The criteria for an extremely severe course were signs of ARF with the need for respiratory support (invasive ventilation), septic shock, and multiple organ failure.

### 2.3. Clinical and Biochemical Surveillances

We obtained the following data for all cases: sex, age, report area, cluster type (family, social, travel, work, community, and vehicle), exposure period, date of disease onset, date of first admission, and date of confirmation. We analyzed the presence of the following risk factors: obesity, arterial hypertension under treatment and risk factors of Charlson Comorbidity Index and their impact on the severity of COVID-19 (age, myocardial Infarction, congestive heart failure, peripheral vascular disease, cerebrovascular disease, dementia, chronic obstructive pulmonary, connective tissue disease, peptic ulcer disease, diabetes mellitus, chronic kidney disease, leukemia, malignant lymphoma, solid tumor, liver disease, and AIDS).

Medical examination of all patients included physical examination and assessment of hemodynamics and respiratory system (respiratory rate, heart rate, blood pressure, SpO_2_, and respiratory distress); calculation of the National Early Warning Score (NEWS), a recommended scoring system for use in COVID-19 patients [12,13]; computed tomography (CT) of the chest with the severity score ranking on a 4-point scale (CT-1, CT-2, CT-3, CT-4); laboratory tests (complete haemogram, basic blood chemistry panel, ferritin test, C-reactive protein, IL-6, lactate dehydrogenase test, D-dimer); ECG; and other instrumental examinations, if required.

### 2.4. Therapy for Patients with COVID-19 Infection

In patients with a mild or moderately severe course of disease, treatment of COVID-19 and its complications included antibacterial and antiviral drugs, prevention of hypercoagulability and disseminated intravascular coagulation, symptom-related treatment, and oxygen therapy. To prevent or treat the cytokine storm depending on the disease severity, in patients with progressive moderately severe or severe disease course, standard treatment was supplemented with convalescent plasma therapy, anticytokine drugs: interleukine-6 (IL-6) inhibitors (tocilizumab, olokizumab, levilimab), IL-1 inhibitors (canakinumab, RH104), JAK inhibitors (tofacitinib, ruxolitinib, baricitinib), Bcr-Abl tyrosine kinase inhibitor (radotinib), and glucocorticoids for some cases. Respiratory therapy, modified antibacterial therapy, extracorporeal membrane oxygenation, sepsis, and septic shock treatment (extracorporeal detoxication and blood purification, etc.) were performed in a stepwise fashion according to indications [9].

### 2.5. Library Preparation and Exome Sequencing

Exome sequencing was performed using either Illumina or MGI sequencing platform. For Illumina, we started gDNA libraries preparation with 500 ng of gDNA sheared using Covaris S2 Focused-ultrasonicator. The fragmented DNA was then converted into DNA-libraries using KAPA HyperPrep Kit (Roche, Basel, Switzerland). The exome-enrichment of DNA-libraries was done using HyperCap Target Enrichment kit and SeqCap EZ Share Choice XL Probes IDP2_REZ clinical exome probe set (Roche, Switzerland), following the manufacturer’s protocol.

For MGISEQ, gDNA libraries preparation started with 500 ng of gDNA sheared using Covaris S2 Focused-ultrasonicator. The fragmented DNA was then converted into DNA-libraries using KAPA HyperPrep Kit (Roche, Switzerland) in combination with MGIEasy DNA Adapters-96 (MGI, Shanghai, China). The exome-enrichment of DNA-libraries was done using HyperCap Target Enrichment kit and IDP2 clinical exome probe set (Roche, Switzerland), according to the manufacturer’s protocol with the following modifications: 1 μL of Block3 and 10 μL of Block4 reagents from the MGIEasy Exome Capture Accessory kit were added to the hybridization mix instead of KAPA Universal Enhancing Oligos, and the final library amplification was done using MGI PCR Primer Mix. To prepare the DNA libraries for sequencing we used MGIEasy Circularization Module V2.0 (MGI, China).

Paired-end reads no shorter than 100bp were generated for each sample.

### 2.6. Variant Calling in Patient Exomes

Each exome sample was aligned onto a GRCh38.p13 reference genome assembly provided in the Genome Analysis ToolKit (GATK) [14] bundle using the BWA MEM read aligner v. 0.7.17 [15]. Next, genetic variants in exome sequencing data were searched using the GATK HaplotypeCaller v.4.1.4 [16]), after which cohort genotyping of the samples was performed. Then, the obtained variants were filtered using GATK: all genotypes with the total read depth less than 10 were set to missing, and then the Variant Quality Score Recalibration (VQSR) was performed with the strict (VQSLOD < 90.0) truth sensitivity thresholds. Filtered variants have been annotated with the Ensembl Variant Effect Predictor v. 103.1 [17].

### 2.7. Phenotype Processing

Prior to association analysis, the phenotypic information was preprocessed in the following way: first, outliers—values that deviate more than 3 standard deviations from the population average—were eliminated, and the data was normalized using the rank-based inverse normal transformation (IRNT) (https://github.com/edm1/rank-based-INT, accessed on 15 June 2021), as normalization of phenotypic data using the inverse IRNT may increase the power of genome-wide association analyses [18]. For several biochemical parameters which were measured on different days since hospitalization, additional measures were calculated to represent the dynamical changes in parameter values. These included the maximum and minimum value of each parameter, the difference between maximum and minimum, the difference between the first day and the last recorded value (prior to discharge or death). For further analysis, features with a coverage of more than 75 percent were selected. Principal components were included in the analysis as aggregation characteristics of the individual’s traits. In total, 28 continuous and 31 categorical features were used for association analysis.

### 2.8. Common Variant Association Analysis (CVAS)

To find associations between genetic variants and disease-related traits we conducted common variant association analysis using the Hail framework for Python v. 0.2.63 (https://hail.is/, accessed on 1 January 2022). Before the analysis, variant- and sample-level quality control was conducted: first, we filtered variants with a call rate (i.e., the proportion of non-missing genotypes) of less than 0.9, the minimal allele frequency of less than 0.05, and the Hardy-Weinberg equilibrium (HWE) *p*-value of less than 0.001. We also filtered out samples with a call rate of less than 0.95 and a heterozygous-to-homozygous call ratio of more than 3. Finally, we performed principal components analysis (PCA) on the resulting set of genetic variants. No outliers were found during PCA.

After quality control, a set of 13,983 high-quality SNPs and 757 samples remained for the association analysis. Association test was conducted using a linear regression method, including several covariates: age, sex, sequencing platform and run, PCA scores for two first principal components (estimated using genetic data), and the Charlson comorbidity index. Association results were evaluated using Manhattan and quantile-quantile (Q-Q) plots, as well as the genomic inflation factor ($\lambda_{GC}$). The plots were drawn using both Hail library and the CMplot package for R (https://github.com/YinLiLin/CMplot, accessed on 15 January 2022).Variants with *p* < 10 ^−4^ were selected as candidate associations for traits that showed the presence of association signal on the Q-Q plots. Following the selection of significant associations, variants were grouped into independent loci by using the clumping method in PLINK v. 1.9 [19].

### 2.9. Rare Variants Associations Studies

In addition to CVAS, we also conducted rare variant association analysis with gene-level and pathway-level aggregation of variant frequencies. Similarly to CVAS, sample- and variant-level quality control was conducted; however, rare variants (MAF < 0.05) were selected during filtering instead of the common ones. Only high-impact variants were selected according to the VEP annotations, including splice acceptor and splice donor site variants, nonsense variants frameshift variants, as well as variants leading to loss of start and stop codons. After quality control and selection of variants, variant counts were aggregated by calculation of the total number of individuals carrying selected variants in each gene. We used Fisher’s exact test to assess the significance of the differences between individuals with different values of binary traits, such as the outcome of hospitalization (death/recovery), severity (severe/not severe), and the presence/absence of a cytokine storm. Results were evaluated using Manhattan and quantile-quantile plots as described above.

### 2.10. Replication of the Identified Associations and Functional Evidence Mining

To prove the biological relevance and significance of the identified variants, we sought for additional evidence of the role of the variants in COVID-19 or other relevant complex traits. For replication of the identified associations, we used the results of the COVID-19 HG project [6] release 6, as well as summary statistics of the Severe COVID-19 GWAS Group study of Spanish and Italian patients [5]. For COVID-19 HG, all four comparisons performed by the study authors were used in replication. We considered all variants with Bonferroni-corrected *p*-value less than 0.05 to be successfully replicated.

In order to identify additional (non-COVID-19 related) associations of the identified variants, we searched for information about their association with other complex traits or gene expression levels using Global Biobank Engine [20] or the Genotype Tissue Expression (GTEx) v8 portal [21], respectively. For Global Biobank Engine, all associations with *p* < 10^−5^ were considered as significant PheWAS hits (the cutoff was chosen as the approximate Bonferroni-corrected significance threshold for phenome-wide analysis in the UK Biobank data).

### 2.11. Construction of the Risk Score

For the validation of the relevance of the identified variants for predicting disease severity and outcome, we calculated a risk score that summarized risk effects from all lead SNPs in each patient. The risk score was calculated as follows:$$s_{j}=\sum_{i=1}^{n}g_{ij}\times \beta_{i},$$
where $s_{j}$ is the score value for patient *j*, *n* is the number of lead SNPs, $g_{ij}$ is the number of risk alleles at variant site *i* in patient *j*, and $\beta_{i}$ is the scaled effect size for each risk allele at variant site *i*.

The score was computed for each patient, and then the cohort was divided into two groups corresponding to the top decile of patients with the highest risk (i.e., 10% of all patients with the highest score values) and the remaining patients in the dataset. After that, Wilcoxon-Mann–Whitney U-test and the chi-squared tests were applied to test for differences in the continuous and categorical features (individual’s death, COVID-19 severity, the presence of a cytokine storm), respectively.

## 3. Results

### 3.1. Study Design and Data Preprocessing

To analyze the genetic susceptibility factors to severe COVID-19, we used clinical exome sequencing data of 840 individuals from a cohort of patients of the City Hospital No. 40 with confirmed COVID-19 diagnosis.

Prior to all further analyses, the sequencing data were uniformly processed (see Methods for more details on the bioinformatic pipeline used) and jointly genotyped. A total of 727,656 genetic variants were discovered in our sample. 98,382 of these variants were non-synonymous variants (including missense and putative loss-of-function (pLoF) variants). After filtering out variants with low quality and/or call rate, 13,983 of the remaining ones were common (AF ≥ 5% and call rate greater than 0.95 in the study sample). Out of the remaining rare (AF < 5%) variants, 1884 variants were annotated as pLoF variants in the canonical transcripts of 1121 protein-coding genes. All individuals were assessed for the presence of monogenic immune system disorders, with no pathogenic variants identified according to the ClinVar database (ClinVar v. 20211130 was used for this analysis).

Identification of significant genome- or exome-wide associations can be difficult in cohorts with limited size; hence, we decided to undertake a more systematic approach and analyze the genetic factors of COVID-19 using an extensive collection of phenotypic data available for our cohort of patients. A broad set of more than 100 quantitative and binary traits were collected for each patient. The set of traits included the major parameters that serve as predictive risk factors of severe COVID-19 according to a recent publications [22]: serum levels of key cytokines such as the C-reactive protein and interleukin-6 (IL-6); levels of ferritin, D-dimer, lactate dehydrogenase (LDH), glucose, and creatine in the serum; blood cell count (lymphocytes, leukocytes, neutrophils per mL of blood sample); lung involvement score derived from CT images, as well as the NEWS score. Most traits were recorded each two days during the course of hospitalization. As expected, the recorded values of most of these traits differed substantially for patients with different outcomes (death or recovery) of hospitalization (Figure 1a; Appendix A) or disease severity (Appendix A). All quantitative traits were additionally pre-processed for further association analysis (Appendix A).

### 3.2. Genome-Wide Association Analysis Using a Deeply Phenotype Cohort


Next, we used the obtained set of phenotypes to search for genes and genetic variants associated with COVID-19 severity. To do so, we applied a multi-perspective analysis strategy by using both common and rare variant association tests (Figure 1b). For binary traits such as death, severity, and the presence or absence of cytokine storm, we performed both common variant association analysis using the linear regression method and the gene-level association test for rare pLoF variants. For quantitative traits, we performed common variant association analysis using the values obtained by the IRNT transformation (see above). The results obtained by each of the analysis approach will be detailed below.

We first conducted common- and rare-variant association analysis with binary traits (death and severity). Common variant association analysis identified no significant associations and no evidence for the exome-wide association signal as shown by the quantile-quantile plots (Appendix A).

We next tested the involvement of rare variants in clinically significant genes by conducting a series of rare variant association tests using both gene- and pathway-level aggregation of variant counts (a strategy similar to the one used by Povysil et al. [23]). To enhance our analysis, we performed both within-cohort tests (i.e., association analysis based on comparison of patients with different COVID-19 outcome or severity) and a comparison with the populational allele frequencies [24]. Concordantly with the results obtained by Povysil et al. [23], we found no genes and pathways showing significant association with disease severity or outcome in our dataset (Appendix A).

We next turned to the analysis of single-variant associations using a broad panel of quantitative trait data. In this analysis, we performed exome-wide association analysis using a set of 13,983 common (MAF > 0.05) variants discovered in our genotype dataset and a set of 53 pre-processed quantitative traits with low missingness rate (for a full list of traits, see Appendix A). After the initial round of GWAS, the results for each trait were manually curated by inspection of the Q-Q plots (a full set of Q-Q plots for all traits is available in the project repository). In total, we found 5 quantitative traits that showed modest exome-wide association signals (Figure 2). These include the serum C-reactive protein (CRP) levels, lymphocyte, leukocyte, and neutrophil counts, and the lung involvement assessed using CT analysis.

In total, 15 variants showed association at $p<10^{-4}$ for the selected quantitative traits. Only two of the identified variants reached exome-wide significance threshold at ($3.5\times 10^{-6}$) (a threshold corresponding to the standard significance level of p<0.05 corrected for the number of variants tested). This variant showed significant associations with both leukocyte and neutrophil counts (this result can be explained by a high degree of correlation between these traits). Clustering of these variants by linkage disequilibrium (LD) identified 11 independent loci (1—for the serum CRP levels; 2—for lymphocyte, leukocyte, and neutrophil count; and 5—for the CT-based lung involvement score; a list of the lead SNPs at each locus is given in Table 1). Four out of these substitutions were located in the coding sequences, while the rest of the variants were intronic or other non-coding variants.

Of the 11 independent variants identified in our analysis, 9 corresponded to significant cis-eQTLs according to the Genotype Tissues Expression (GTEx) data. Four of these variants corresponded to cis-eQTLs affecting the expression of multiple genes across multiple tissues. Of these, three variants had the most significant effect on neighboring genes: the rs2276638 intron variant in the *DNAJB2* gene had the most significant effect on the expression of the *PTPRN* gene in whole blood according to the GTEx data ($p=2\times 10^{-27}$); the rs33985936 variant in *SCN11A* had the highest effect on the expression of the *TTC21A* gene in esophagus; and the rs112544 variant in *LZTR1* had the most significant and broad impact on the expression of the *THAP7-AS1* antisense transcript. Of the remaining five variants with significant cis-eQTL signal, four had a significant effect on the expression of the gene bearing the corresponding variant, and only one affected the expression of the neighboring gene. Taken together, these results do not allow to draw a confident conclusion regarding the exact target gene influencing the phenotyping for the majority of variants; however, it appears likely that the variants in *ATXN1*, *PKHD1*, *SETX*, *PIEZO1*, and *CDH23* have a direct impact on the phenotype by changing the function (in case of missense variants in *ATXN1* and *CDH23*) or expression levels of the corresponding gene.

### 3.3. Replication and Validation of the Identified Markers

While we identified 11 independent genetic variants that are associated with quantitative traits that are directly connected to the disease severity and outcome, it is important to note that the significance level of these associations is not sufficient for making a confident conclusion about the effects on the patient phenotype. This predicates the need for additional replication of the observed associations and validation of their true role in the pathogenesis of COVID-19.

To obtain such a validation, we first questioned whether the identified variants can be used to directly predict the severity of the disease and/or outcome in our cohort. We began by constructing a simple risk score by computing the weighted sum of risk alleles in the genotype of each patient (see Methods for details). The score had a nearly normal distribution (Figure 3a). To test whether such a score has a power to predict the severity or outcome of the hospitalization in COVID-19 patients, we then selected the patients belonging to the top decile of the score distribution (i.e., 10% of all patients with the highest score values). We then used chi-squared statistics to compare disease severity and outcome in these patients and the rest of our sample. Indeed, we found significant differences in all comparisons (Figure 3b), with patients belonging to the top risk score decile having greater probability of death and greater disease severity. Concordantly with this analysis, logistic regression based on the 11 identified markers predicts COVID-19 outcome with ROC/AUC = 0.59. These results confirms that the identified variants can be considered as genetic risk factors of severe COVID-19.

Having demonstrated the general relevance of the identified variants for predicting the severity of the disease and its outcome, we then analyzed the enrichment of gene sets for the identified loci to test for common function among genes harboring top associations. We performed such an enrichment analysis using several tools designed for GWAS data, including LSEA [25] and FUMA [26]. Unfortunately, no significant enrichment of molecular pathways was identified for any of the individual traits and for the combined set of 11 loci (data not shown).

We next went on to replicate the observed associations in independent studies [5,6]. The results of the replication are presented in Table 2. When using the COVID-19 HG data, we successfully replicated only one of 11 candidate associations (rs33985936 in *SCN11A*) which showed modest significance in the analysis of COVID-19 patients vs. population (C2 comparison in COVID-19 HG). We also attempted replicating our findings in the results of the Severe COVID-19 GWAS Group (a study involving patients and controls of Spanish and Italian ancestry). No genetic variants were successfully replicated in this study (Table 2).

In addition to replicating the associations in other studies of the COVD-19 host genetics, we sought to identify other (not COVID-19-related) complex traits associated with the identified variants, To this end, we performed phenome-wide association analysis (PheWAS) using the Global Biobank Engine [20]. All associations at $p<10^{-5}$ were considered as significant PheWAS hits. We were able to identify PheWAS hits for 3 out of 11 tested variants. Remarkably, all identified phenome-wide associations corresponded to missense variants and were identified for blood-related traits. The rs33985936 variant in the *SCN11A* gene, the only variant that was replicated in the COVID-19 HG cohort, showed significant association with platelet crit and platelet count in the UK Biobank data. In addition to this variant, rs16885 in *ATXN1* showed significant PheWAS association with mean corpuscular hemoglobin levels, and the rs4747194 variant in *CDH23* was connected to the percentage of monocytes in the blood. These results provide additional support for the biological role of the identified missense variants in driving COVID-19 related traits.

To sum up, we identified a set of 11 genetic variants showing modest association with quantitative and nominal traits linked to COVID-19 severity. For three of these variants, we were able to find supporting evidence substantiating their role in the COVID-19 pathogenesis.

## 4. Discussion

The COVID-19 pandemic has drawn significant attention to the interactions between the host genome and pathogens in infectious disease pathogenesis. Over the last two years, a series of publications addressed the issue of hereditary predisposition to SARS-CoV-2 infection and severe disease [3]. Analysis of genetic associations in cohorts of limited size, especially when no genome-wide genotypes are available, might also hinder the discovery of new susceptibility loci in underrepresented populations. Hence, sophisticated approaches have to be used to tackle the low statistical power of the analysis. In this work, we have utilized a wide variety of quantitative traits recorded in a cohort of Russian COVID-19 patients to identify novel loci with a possible influence on disease severity and outcome. The results of our analysis demonstrate that such an approach can be useful to identify sets of genetic variants that have a modest power to discriminate between patients with different levels of COVID-19 severity (Figure 2 and Figure 3). This observation further confirms the importance of the basic predictive risk factors of cytokine storm in COVID-19 which we have described previously [22].

Only one of the identified variants successfully passed replication in external cohorts, with two additional variants demonstrating nominal significance in independent studies. These results are similar to the ones obtained by Li et al. [27]. As argued previously, a low replication rate might reflect both differences in study design and differences between populations. Perhaps more importantly, we observed significant phenome-wide associations for three of our variants in the UK Biobank data. Importantly, all of the PheWAS hits corresponded to blood-related traits (Table 2), supporting the relevance of the identified associations. Furthermore, the analysis of GTEx eQTLs also shows that many of the identified variants affect the expression of genes in immune cells or in whole blood (e.g., rs2276638 in *DNAJB2*, rs34600315 in *PIEZO1*). Few of the identified variants affect gene expression in the lungs. This observation may be explained by the specifics of the analysis strategy which is mostly focused on blood-borne traits in COVID-19 patients.

Several genes belonging to the top associated loci in our study deserve a detailed discussion. First, the most significant (and the only significant on the exome-wide level) variant corresponded to the *MYH14* gene encoding for nonmuscle myosin II C (NMIIC) predominantly expressed in the inner ear, including the organ of Corti [28]. Non-synonymous variants in *MYH14* increase risk of neurological progression of type 2 diabetes and peripheral neuropathy [29,30]. Such variants exhibit a dominant-negative effect by inhibiting the division of mitochondria [29]. Hence, alterations in the *MYH14* function may thereby increasing the severity of respiratory failure in COVID-19 infection. However, further investigations are needed to establish the exact role of *MYH14* in disease progression.

Second, the *DNAJB2* gene encodes an important protein of the Hsp40 chaperone group. Such proteins are known to contribute to the substrate-specificity of other chaperones and mediate the stress response [31,32]. The involvement of the *DNAJB2* gene variation in the levels of lung damage upon SARS-CoV-2 infection is interesting and may point to the role played by the stress response pathways and protein quality control in disease severity. it can be hypothesized that the heat stress response is important for alleviation of negative effects of inflammation on the structure of the tissue; thus, deviation in the regulation of response to heat dresses may trigger the destruction of the lung tissue and cause lung fibrosis and respiratory problems in COVID-19 patients.

The association of the locus spanning the *LZTR1* gene is also notable as this gene encodes an important leucine-zipper transcriptional regulator that is linked to cell proliferation in cancer [33]. The lead variant at the locus, rs112544, has a significant and broad impact on the expression of the *THAP7* gene and its antisense transcript, *THAP7-AS1*. *THAP7* encodes a transcriptional repressor that acts via histone deacetylation [34]). Notably, *THAP7* overexpression induces permissiveness of human hepatoma Huh7 cells to hepatitis C viral invasion [35]. These data suggest that both *LZTR1 per se*, as target genes affected by rs112544, may be involved in immune system function and anti-viral response. *PIEZO1* codes for mechanically-gated ion channels with multiple roles in human organisms, mutations in this gene are comorbid with pathological conditions, namely, hereditary anemia [36], congenital lymphedema [37], lymphatic dysplasia [38], and others, implying that alterations in *PIEZO1* may affect the immune response to SARS-CoV-2.

Variants in fibrocystin (encoded by the *PKHD1* gene) worsen the ramifications of the autosomal recessive polycystic kidney disease (ARPKD) in children and adults [39]. Notably, the gene’s polymorphisms were found to be associated with mild cognitive impairment [40] and metachronous liver cancer [41]. Similarly, other genes located at the 11 identified loci (e.g., *ATXN1*, *GABBR2*, *SETX*, *CDH23*) are also implicated in nervous system pathology but are not clearly linked to immunity and/or infectious disease response [42,43,44,45,46]. This result might indicate a certain relationship between nervous system function and COVID-19 severity; however, it is important to note that no significant overrepresentation of nervous system genes was identified at the associated loci.

It has to be noted that the overall strength of the observed associations in our study is moderate, and only 1 of the loci pass replication in the independent cohorts. This observation may be attributed to either weak association signal in our study or population-specific effects of the variants. The low replication rate is also expected given differences in study designs. Moreover, the majority of the identified associations correspond to non-coding (i.e., intronic) variants, suggesting that the actual causal variants might be located further from covered exome regions. However, our results demonstrate the utility of deep laboratory phenotyping of COVID-19 patients for the identification of novel genetic variants affecting the severity and/or outcome of the disease. Hence, we believe that our work may serve as an example of successful indirect identification of genetic risk factors of severe COVID-19.

## Figures and Tables

**Figure 1 genes-13-00534-f001:**
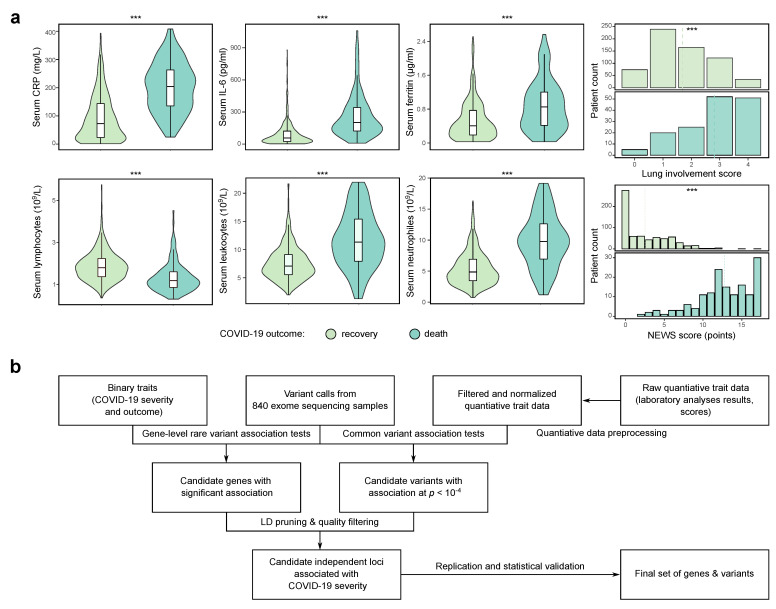
Identification of candidate genetic markers of severe COVID-19 using a deeply phenotype cohort. (**a**) Distributions of selected quantitative traits for individuals with different disease outcome (death or recovery) in the cohort of 840 COVID-19 patients from Russia. Shown are the distributions of the serum C-reactive protein (CRP), interleukin-6, and D-dimer levels, CT-based lung involvement score (ranging from 0 to 4), counts of lymphocytes, leukocytes, and neutrophils in the blood samples, as well as the National Early Warning Score (NEWS). All values shown correspond to maximum values recorded during the course of hospitalization. Values exceeding three standard deviations from the population mean are omitted. ***—p<0.001 in Wilcoxon-Mann-Whitney test (for quantitative traits) or chi-squared test (for categorical traits). (**b**) A schematic representation of the data analysis pipeline employed in the present study.

**Figure 2 genes-13-00534-f002:**
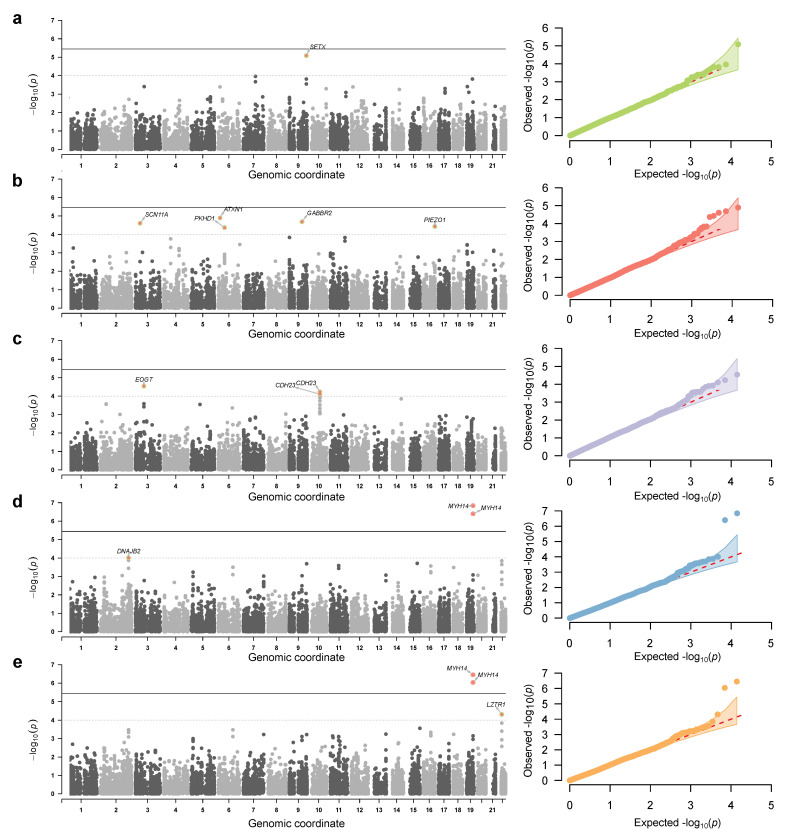
Genome-wide association results for selected quantitative traits in COVID-19 patients. Shown are Manhattan (**left**) and quantile-quantile (**right**) plots of association *p*-values for (from top to bottom) the serum CRP levels (**a**), CT-based lung involvement score (**b**), serum lymphocyte (**c**), leukocyte (**d**), and neutrophil (**e**) counts. Thresholds on the Manhattan plots correspond to the exome-wide significance cutoff ($4\times 10^{-6}$) and the sub looser $p=10^{-4}$ cutoff used to select candidate associations.

**Figure 3 genes-13-00534-f003:**
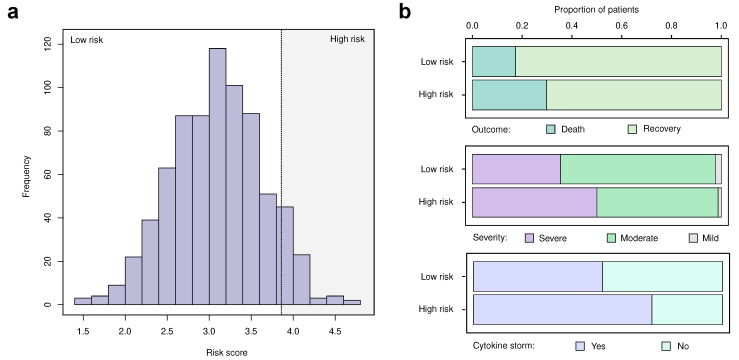
A risk score based on 11 identified variants predicts disease severity and outcome. (**a**) Distribution of the risk score computed based on the 11 lead SNPs shown in Table 1. Shaded area indicates the top score decile corresponding to high-risk individuals. (**b**) Bar plots showing the proportion of patients with different outcome (top), severity (middle), or presence of the cytokine storm (bottom) in the high-risk and low-risk groups. In all cases, p<0.05 in chi-squared test.

**Table 1 genes-13-00534-t001:** Candidate genetic variants associated with COVID-19 related quantitative traits in a cohort of Russian patients.

Locus	rsID	Substitution	AF *	Trait(s)	Gene	Consequence	β **	*p*-Value	GTEx eQTLs ***
2:219280564	rs2276638	6247C>G	0.135	Leukocytes	*DNAJB2*	intron variant	−0.29	$9.84\times 10^{-5}$	Multiple genes and tissues
3:38894643	rs33985936	c.2725G>T (p.Val909Phe)	0.241	CT score	*SCN11A*	missense variant	−0.24	$2.50\times 10^{-5}$	Multiple genes and tissues
3:68997990	rs4855544	g.20905C>A	0.332	Lymphocytes	*EOGT*	intron variant	0.23	$2.88\times 10^{-5}$	Multiple genes and tissues
6:16306520	rs16885	c.2257C>T (p.Pro753Ala)	0.197	CT score	*ATXN1*	missense variant	0.27	$1.28\times 10^{-5}$	none
6:51830849	rs1571084	g.261777T>A	0.333	CT score	*PKHD1*	intron variant	0.21	$4.30\times 10^{-5}$	*PKHD1* (skin)
9:98299383	rs41273925	g.414815C>G	0.081	CT score	*GABBR2*	intron variant	0.38	$2.06\times 10^{-5}$	*TBC1D2* (thyroid)
9:132278286	rs11243705	g.81700A>G	0.180	CRP	*SETX*	intron variant	0.30	$8.18\times 10^{-6}$	*SETX* (multi-tissue)
10:71799129	rs4747194	c.7073G>T (p.Arg2358Gln)	0.243	Lymphocytes	*CDH23*	missense variant	0.25	$5.84\times 10^{-5}$	*CDH23* (colon, testis), *PSAP* (multi-tissue)
16:88738516	rs34600315	c.*648_*649del	0.657	CT score	*PIEZO1*	non coding transcript exon variant	0.21	$3.73\times 10^{-5}$	*PIEZO1* (whole blood)
19:50259161	rs1651553	c.2127A>G	0.770	Leukocytes, neutrophiles	*MYH14*	synonymous variant	0.32, 0.31	$1.45\times 10^{-7}$, $3.55\times 10^{-7}$	none
22:20992196	rs112544	g.14928T>G	0.709	Neutrophiles	*LZTR1*	intron variant	0.23	$4.88\times 10^{-5}$	Multiple genes and tissues

*—allele frequency is given with respect to the non-reference allele; **—the effect sizes are given with respect to the IRNT-transformed values of quantitative traits; ***—data for the
GTEx Analysis Release v8 (full list of significant cis-eQTLs is available in Appendix A). Bold font indicates a *p*-value passing formal exome-wide significance threshold.
*DNAJB*2—Dna J heat shock protein family (Hsp40) member B2; *SCN11A*—sodium voltage-gated channel alpha subunit 11; *EOGT*—EGF domain specific O-linked N-acetylglucosamine
transferase; *ATXN1*—ataxin 1; *PKHD1*—PKHD1 ciliary IPT domain containing fibrocystin/polyductin; *GABBR2*—gamma-aminobutyric acid type B receptor subunit 2; *SETX*—senataxin;
*CDH23*—cadherin related 23; *PIEZO1*—piezo type mechanosensitive ion channel component 1; *MYH14*—myosin heavy chain 14; *LZTR1*—leucine zipper like transcription regulator 1.

**Table 2 genes-13-00534-t002:** Replication of the association for the 11 identified variants in independent cohorts.

Variant	Gene	*p*-Value (This Work)	A2 ^†,^*	B1 ^†,^**	B2 ^†,^***	C2 ^†,****^	The Severe COVID-19 GWAS Group ^††^	UK Biobank PheWAS Traits ^†††^
rs2276638	*DNAJB2*	$9.84\times 10^{-5}$	0.424	0.835	0.808	0.377	0.6281	none
rs33985936	*SCN11A*	$2.50\times 10^{-5}$	0.738	0.516	0.108	0.001	0.3382	Platelet count, platelet crit
rs4855544	*EOGT*	$2.88\times 10^{-5}$	0.118	0.611	0.65	0.098	0.6478	none
rs16885	*ATXN1*	$1.28\times 10^{-5}$	0.998	0.988	0.293	0.015	0.1201	Mean corpuscular hemoglobin
rs1571084	*PKHD1*	$4.30\times 10^{-5}$	0.075	0.269	0.836	0.906	0.8684	none
rs41273925	*GABBR2*	$2.06\times 10^{-5}$	0.904	0.298	0.701	0.034	0.9353	none
rs11243705	*SETX*	$8.18\times 10^{-6}$	0.116	0.948	0.995	0.613	0.857	none
rs4747194	*CDH23*	$5.84\times 10^{-5}$	0.133	0.514	0.718	0.463	0.9983	Monocyte %
rs34600315	*PIEZO1*	$3.73\times 10^{-5}$	n.a.	0.699	0.976	0.770	0.3098	none
rs1651553	*MYH14*	$1.45\times 10^{-7}$	0.690	0.068	0.539	0.329	0.8383	none
rs112544	*LZTR1*	$4.88\times 10^{-5}$	0.273	0.618	0.359	0.524	0.9679	none

Bold font corresponds to variants passing replication at adjusted *p*-value < 0.05. ^†^—COVID-19 HG; ^††^—COVID-
19 cases vs. controls from Italy and Spain (corrected for 10 genomics PCs, sex, and age); ^†††^—phenome-wide
associations were selected using the Global Biobank Engine at *p*-value 10^−5^; *—very severe respiratory confirmed
COVID-19 vs. population; **—hospitalized COVID-19 vs. not hospitalized COVID-19; ***—hospitalized
COVID-19 vs. population; ****—COVID-19 vs. population.

## Data Availability

All data and code pertinent to the results presented in this work are available at https://github.com/bioinf/covid19-exome/.

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
