# Peer review of "Identification of Genetic Risk Factors of Severe COVID-19 Using Extensive Phenotypic Data: A Proof-of-Concept Study in a Cohort of Russian Patients"

_genes, 2022, doi:10.3390/genes13030534_

Round 1

Reviewer 1 Report

I think this paper is very interesting because it shows the association between the phenotyping of COVID-19 infection, which is one of the most interesting topics for medical professionals in recent years, and the genetic variants. 

I would like to ask you to reconsider one of the following points.

#1 I found the description of the functional basis for the effects of MYH14 variants to be a bit unclear. The citation I would like you to be a little more specific about the "it may be hypothesized that the effects of MYH14 variants might be indirect and caused by other genes in the same locus."

Author Response

However, it should be noted that the MYH14 gene variant (R941L) exhibits a dominant-negative effect by inhibiting the division of mitochondria [Almutawa W, 2019] - the energy and respiratory structures of the cells, thereby increasing the severity of respiratory failure in COVID-19 infection. In addition, the MYH14 gene variant (R941L) is associated with the development of peripheral neuropathy, which can be a symptom of COVID-19 infection [Almutawa W, 2019, Diamond KB, 2021].

Almutawa W, Christopher Smith , Rasha Sabouny, Ryan B Smit , Tian Zhao, Rachel Wong, Laurie Lee-Glover, Justine Desrochers-Goyette, Hema Saranya Ilamathi, Care4Rare Canada Consortium; Oksana Suchowersky, Marc Germain, Paul E Mains, Jillian S Parboosingh, Gerald Pfeffer, A Micheil Innes, Timothy E Shutt. The R941L mutation in MYH14 disrupts mitochondrial fission and associates with peripheral neuropathy. EBioMedicine. 2019;45:379-392. doi: 10.1016/j.ebiom.2019.06.018.

Diamond KB, Weisberg M D, Ng M.K., Erez O, Edelstein D. COVID-19 Peripheral Neuropathy: A Report of Three Cases. Cureus. 2021;13(9): e18132. doi: 10.7759/cureus.18132

Reviewer 2 Report

The manuscript by Sergey G. Shcherbak and cols. show interesting research to identify the genetic risk factors that predict COVID-19 severity and outcome by searching for candidate genes and genetic variants associated with a variety of quantitative and binary traits in a cohort of 840 COVID-19 patients from Russia.

Although the authors found no gene- or pathway-level associations with the disease severity and outcome, they discovered 11 candidate loci associated with quantitative traits in COVID-19 patients. Still, only one of the identified variants was successfully replicated in an independent study. According to UK Biobank data, three of the variants were associated with blood-related quantitative traits.

Some minor comments.

The study design looks well done, and the sample size is extensive; however, the authors should state about the power-sample size and discuss, briefly.

Most of the identified SNPs are intronic variants; please discuss this consequence.

In the results section, authors are repetitive with the methods section; please avoid constantly referring to the methods section at least being extremely necessary.

Figure 1a, please re-order the tags: recovery/death instead of death/recovery, just for looking at the figure.

The introduction and discussion sections are redundant; please reformulate both. You can use some recently published reviews in genomic/pharmacogenomic to avoid describing each gene, particularly in the introduction.

Author Response

The manuscript by Sergey G. Shcherbak and cols. show interesting research to identify the genetic risk factors that predict COVID-19 severity and outcome by searching for candidate genes and genetic variants associated with a variety of quantitative and binary traits in a cohort of 840 COVID-19 patients from Russia.

Although the authors found no gene- or pathway-level associations with the disease severity and outcome, they discovered 11 candidate loci associated with quantitative traits in COVID-19 patients. Still, only one of the identified variants was successfully replicated in an independent study. According to UK Biobank data, three of the variants were associated with blood-related quantitative traits.

Authors: We thank the Reviewer for the positive assessment of our work.

Some minor comments.

The study design looks well done, and the sample size is extensive; however, the authors should state about the power-sample size and discuss, briefly.

Authors: The power analysis in our case is complicated by a number of factors that include specificity of the sequencing kit and heterogeneity of the traits that were used in the analysis.. Nevertheless, recent studies suggest that the sample of 800 individuals optimizes statistical power under certain reasonable assumptions regarding quantitative trait heritability (e.g., Wang and Xu, 2019). We have added this information to the Methods section of the manuscript.

Most of the identified SNPs are intronic variants; please discuss this consequence.

Authors:  The corresponding discussion was added to the manuscript (p. 13, lines 435-440).

In the results section, authors are repetitive with the methods section; please avoid constantly referring to the methods section at least being extremely necessary.

Authors: We revised the Results and Methods sections accordingly; excessive references to Methods were omitted.

Figure 1a, please re-order the tags: recovery/death instead of death/recovery, just for looking at the figure.

Authors: The groups were re-ordered.

The introduction and discussion sections are redundant; please reformulate both. You can use some recently published reviews in genomic/pharmacogenomic to avoid describing each gene, particularly in the introduction.

Authors: These sections were changed to avoid redundancy.